# Anthropomorphic Cyber-Physical Systems

**Evgenii Vityaev**
Sobolev institute of mathematics SB RAS, Novosibirsk, Russia
`vityaev@math.nsc.ru`

## Abstract

The concept of Cyber-Physical Systems (CPS) is quite broad. Here is one of the most comprehensive definitions. The main operating principle of Cyber-Physical Systems is the deep integration between physical and computational elements. Cyber-physical systems obtain data from sensors in the real world, analyze this data, and use it to further control physical elements. Due to such interaction, a cyber-physical system is capable of operating effectively in changing conditions, analogous to the human organism, in order to develop precisely the product it currently needs. To understand how such deep integration of control commands, physical system elements, sensors, and reality is possible to achieve the desired effect in real-time, we turn to a thoroughly developed physiological theory of purposeful behavior in humans and animals — the Theory of Functional Systems of brain operation. We present the formal model of functional systems and describe how it works in purposeful behavior in humans. Based on this model, we further develop Anthropomorphic Cyber-Physical Systems.

## 1 Introduction

The concept of Cyber-Physical Systems (CPS) is quite broad. Cyber-physical systems obtain data from sensors in the real world, analyze this data, and use it to further control physical elements. Cyber-physical systems are capable of operating effectively in changing conditions, analogous to the human organism, in order to develop precisely the product it currently needs. Moreover, the cycle

"control — data acquisition — data processing — control"

in a well-established CPS should yield positive results each time. We will assume that control and positive result have some target parameters.

To understand how such deep integration of control commands, physical system elements, sensors, and reality is possible to achieve the desired effect in real-time, we turn to a thoroughly developed physiological theory of purposeful behavior in humans and animals — the Theory of Functional Systems of brain operation (Anokhin, 1978; Vityaev, 2015).

In works (Anokhin, 1978; Vityaev, 2015; 2014; Vityaev et al., 2023; 2025) we presented a formalization of TFS as a cognitive architecture of purposeful human activity. Here is a description of purposeful activity in accordance with TFS.

1. When a need (task) arises, motivational arousal arises, which is transmitted to the cerebral cortex and "extracts from memory" all ways to achieve the goal that satisfy this need. Together with each of the ways, the entire sequence and hierarchy of intermediate sub-goals and sub-results that must be achieved in order to achieve the goal is also extracted.

2. For each sub-goal and sub-result, the criterion for achieving this sub-goal and obtaining this sub-result is fixed as a "set of afferent stimuli", indicating that the sub-goal has been achieved and the result has been obtained. This sequence, together with the required parameters of all intermediate and final results, is in a certain sense the control points of the goal achievement process.

3. Only the experience that takes into account the current situation and is applicable to it is extracted from memory. "Trigger" stimuli are also extracted, which indicate the start time of certain actions.

4. For all ways to achieve the goal, the emotion apparatus evaluates the probability of achieving them and the costs associated with labor intensity, risks and possible obstacles. These estimates can be modeled fairly accurately by some utility function.

5. At the decision-making stage, taking into account both the probability of achieving the goal and the costs, only one of the ways is chosen.

6. After that, the goal is achieved by completing the entire sequence and hierarchy of intermediate sub-goals and obtaining all the sub-results along with monitoring their receipt.

7. If all intermediate results and the final result are achieved, then the purposeful behavioral act ends with the last sanctioning stage – reinforcement, when the received method of achieving the goal, taking into account the new situation and possible modifications, is recorded in memory.

8. If any of the intermediate results or the final result is not obtained, then the plan for achieving the goal (sub-goal) is reviewed taking into account other ways to achieve the goal (sub-goal).

According to the TFS description, the cyber-physical control system can be formulated as follows:

1. States space of CPS is defined in terms of corresponding predicates that uniquely determine the state of CPS. These predicates determine both the internal state of the system and the state of the external environment.

2. Control in a cyber-physical system and the required positive result have certain target parameters (Goal) that must be achieved. Changing the target parameters over time with the required trend should be included in the definition of these parameters.

3. CPS may have a series of control points that control the operation of individual parts (sub-systems) of the system. For these control points as sub-goals, target parameters (Goals) are also set in terms of those features that characterize the required operation of these parts.

4. Set of actions is defined together with the definition of how each action transitions the system from one state to another. By an action we also mean a sequence of actions that also transition the system from one state to another.

5. Target parameters of the current state of CPS and its subsystems determine the current Goal states of CPS and sub-goals of its subsystems that include these Target parameters (corresponding predicates are true in these states).

6. Each Goal and sub-goal state retrieves from the memory all probabilistic rules for actions that transition CPS from the current state to this Goal and sub-goal states. If there are no such actions in the memory, then a search of actions is initiated that change the current state of CPS to the Goal and sub-goal states. Thus the set of transitions with corresponding probabilistic rules can be found for this Goal state and sub-goals.

7. Every transition from the current state to the Goal and sub-goals states has not only its probability, defined by corresponding probabilistic rule(s), but also a cost of that transition. Humans have a special emotional feeling about that cost, but CPS may estimate this cost by the corresponding utility function that uses the probability(ies) of the rule(s).

8. Decision-making on choosing the best transition from the current state to the Goal and sub-goals states may be made by taking those transitions that have the greatest value of utility. The utility of the CPS transition to the Goal is a sum of the sub-goals transitions utilities and the Goal.

9. Then the Goal may be achieved by execution of all the actions of those transitions.

10. The CPS has memory. Each really executed action, together with the found sequence of rules, actions and states that led to the Goal achievement satisfying target parameters, is memorized. Unsuccessful control action that did not lead to the Goal achievement is also memorized.

11. The CPS learns. The accumulated set of executed actions in the memory is analyzed, and the entire set R of probabilistic cause-effect rules is extracted from it.

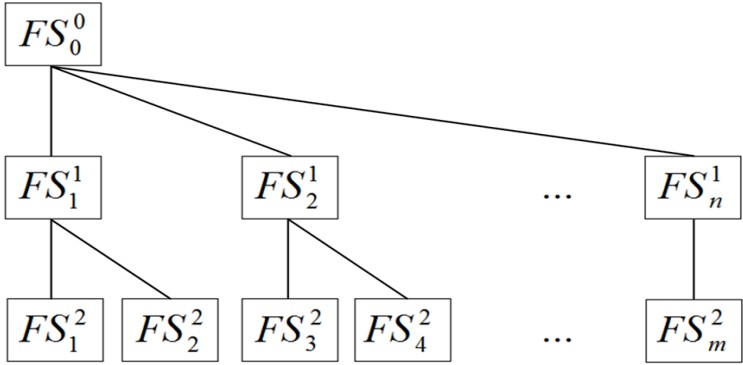

Figure 1: Hierarchy of functional systems

12. Goals prediction in CPS. Prediction of the target parameters for all Goals may be fulfilled by the set of rules R in real time. If any forecasts are unsatisfactory, then these target parameters may be updated and the decision-making process automatically chooses another best transition for them. This may control the CPS process.

13. The CPS has stopping criteria, fixed in constant parameters, for when either the target parameters are no longer improving, or time or resources are exhausted.

## 2 CPS FORMAL MODEL

We present a general model of CPS, focusing on their architecture and learning mechanism. This framework build upon the established physiological Theory of Functional Systems that explain purposeful human behavior Anokhin (1978).

Consider a CPS that works in discrete time $t = 0, 1, \ldots$ and possesses sensors $S_1, \ldots, S_n$ monitoring both internal states and the external environment. Each sensor $S_i$ has possible values $VS_i$. The CPS action repertoire $A_1, \ldots, A_m$ enables environmental interactions. Actions executed at $t_i$ may alter the environment by $t_i + 1$, subsequently changing sensor values.

Since perception occurs exclusively through sensors, the CPS state at time $t$ is represented as a sensor value vector $V(t) = (v_1, \ldots, v_n)$, where $v_i \in VS_i$ denotes the $i$-th sensor's value. Identical sensor vectors yield indistinguishable states. The complete state space for CPS is $SS = (VS_1 \times VS_2 \times \ldots \times VS_n)$.

Given inherent sensor limitations (sensitivity, range, etc.), executing action $A$ in state $S$ may transition this state into multiple possible states. Thus, action $A_i$ functions as:

$$A_i : (SS_i) \rightarrow (SS \times P),$$

where $SS_i \subseteq SS$ contains states where $A_i$ is executable, and $SS \times P$ comprises pairs $(ss, p)$ with $ss \in SS^g$ as final state and $p \in [0, 1]$ as transition probability from initial state $ss \in SS_i$.

We define an event $e^g = (ss_0, ss_e, A_0)$ as a state transition $ss_0 \in SS_0 \rightarrow ss_e \in SS$ via action $A_0$. History $H$ comprises timestamped events $(e, t)$.

Transitioning to a discrete model: For CPS, we define predicates $PS = \{PS_1, \ldots, PS_k\}$ evaluated against sensor values $V$. The CPS state then becomes $S = (ps_1, \ldots, ps_k)$ (boolean predicate values). Crucially, predicate sets $PS$ overlap across agents, enabling shared environmental interpretation.

Predicates may be computed by Neural Networks or other AI systems. Our hybrid Task-Based AI approach treats these as oracles responding to predicate-evaluation requests.

CPS Goal $G$ is a specific set $PS$ of predicates the truth of which in some state unambiguously indicates that the Goal is achieved

$$G = \{ps_1, \ldots, ps_l\}, \quad l \leq k.$$

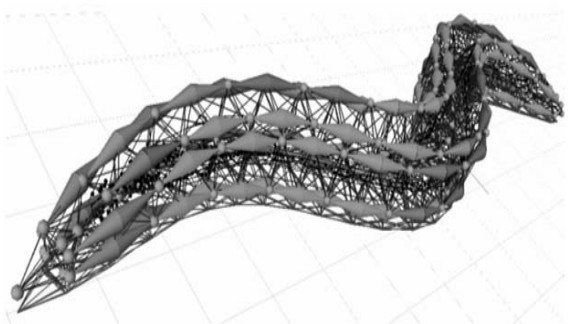

Figure 2: Scheme of the model

In this discrete framework, event $e = (S_0, \ S_e, \ A)$ represents transition $S_0 = (ps_{01}, \ ..., \ ps_{0k}) \rightarrow S_e = (ps_{e1}, \ ..., \ ps_{ek})$ via $A$, with history $H$ remaining $(e, \ t)$ pairs.

Achieving the goal $G$ can include both individual actions in accordance with the corresponding probabilistic rules, and sequences of actions together with the corresponding groups of rules that can be performed for some substructures. In accordance with the theory of functional systems, this is formalized by a functional system that includes both separate rules for actions and functional subsystems for sequences of actions.

CPS achieves goals through the following functional systems.

$$FS^{rank} = (G^{rank}, \ R_1, \ ..., \ R_v, \ FS_1^{rank+1}, \ ... , \ FS_w^{rank+1}), \ w \leq v$$

where $G^{rank}$ is $FS^{rank}$'s primary goal; $R_i$ are rules for individual actions; $FS_1^{rank+1}, \ldots, FS_w^{rank+1}$ are subordinate sequences of actions satisfying $R_1, ..., R_v$ preconditions.

A rule $R$ is a transformation $S_0 \xrightarrow[p]{A} S_e$ where:

- $S_0$: Initial state $(ps_{01}, \ldots, ps_{0n})$
- $S_e$: Final state $(ps_{e1}, \ldots, ps_{en})$ (containing $G^{rank}$ for $FS^{rank}$ rules)
- $A$: Transition action(s)
- $p$: Transition probability

Rule probability $p \in [0, 1]$ is computed as $p = b/a$, where $a$ counts $S_0$ occurrences and $b$ counts $S_0 \rightarrow S_e$ transitions via $A$. Note: Rule probabilities ($p$) differ from environmental transition probabilities ($P$). The FS objective is minimizing their discrepancy through experience.

$FS^{rank}$ may instantiate subordinate subsystems $FS^{rank+1}$ to achieve sub-goals $G^{rank+1}$ that enable rules $G^{rank+1} \xrightarrow[p]{A} G^{rank}$ (Fig. 1). Historical events with outcome $G^{rank+1}$ construct $FS^{rank+1}$'s rule set via semantic probabilistic inference Vityaev (2014).

When $FS^{rank}$ (rank=0,1,...) receives goal $G^{rank}$, it either:

1. Selects highest-probability rule $R_i^{rank}$, OR
2. Delegates to subordinate $FS^{rank+1}$ if: no applicable $R_i^{rank}$ exists for $G^{rank}$, OR applicable $R_i^{rank}$'s probability is lower than the compound probability of achieving $G^{rank+1}$ via subordinates $FS^{rank+k}$ ($k > 0$) and executing $R_i^{rank}$

Upon delegation, $FS^{rank+1}$ attempts $G^{rank+1}$. If successful, $FS^{rank}$ executes its highest-probability rule for goal $G^{rank}$ achievement and executes the corresponding action(s). Failure returns "goal not achieved".

After action(s) $A$ execution, the new state is compared against $G^{rank}$. Successful outcomes reinforce the rule (positive statistics update); failures punish it (negative update). Rule refinement follows semantic probabilistic inference Vityaev (2014).

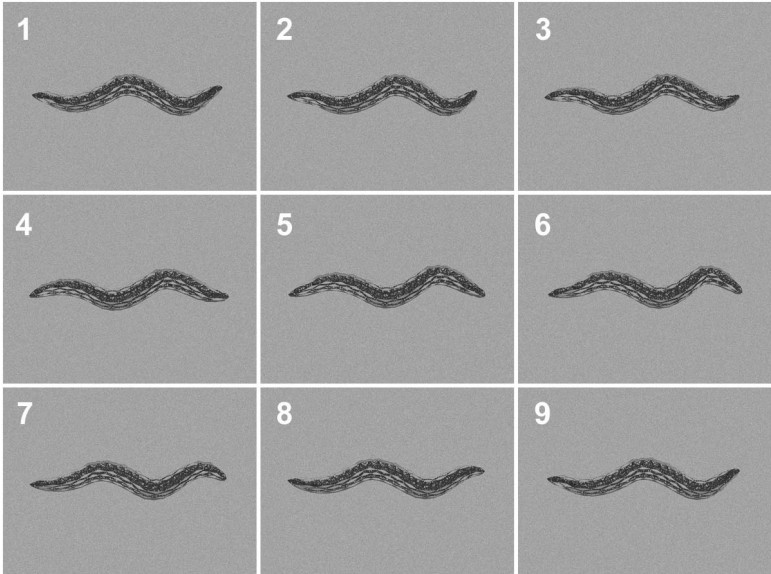

Figure 3: Scheme of the model

## 3 CONCLUSION

This model of CPS was successfully applied to controlling locomotion of the C.elegans nematode Demin & Vityaev (2014). A 3D realistic simulator (Fig.2) of the nematode was used to learn this model. The control system stably learned an effective way of movement forward in 100 working cycles of learning. A considerable visual likeness was observed between the behavior of the model and the behavior of a real nematode (Fig. 3).

ACKNOWLEDGMENTS

The work was financially supported by the State Assignment of the Sobolev Institute of mathematics SB RAS for 2026-2029 "Theory of computability and logical aspects, logical calculus and their semantics". Supervisor: S.S. Goncharov. Project Number: FWNF-2026-0032.

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
