# OpenReview forum: "ANTHROPOMORPHIC CYBER-PHYSICAL SYSTEMS"
_mathai.club/MathAI/2026/Conference — 2026 Oral_

### Official Review · Reviewer_FPKD · 2026-03-12
**ANTHROPOMORPHIC CYBER-PHYSICAL SYSTEMS**

**Rating:** 4
**Confidence:** 4

**Review:**

This manuscript draws an analogy between biological purposeful behavior (as described by Anokhin's Theory of Functional Systems) and cyber-physical systems. The authors argue that TFS provides a well-developed framework for understanding how organisms achieve goals through perception, action, and learning, and that this framework can be adapted to design more adaptive and robust CPS. They present a step-by-step mapping of TFS concepts (need, goal, memory, decision-making, reinforcement) onto CPS components, resulting in an "anthropomorphic" CPS architecture. The paper concludes with a brief reference to a previous application: controlling locomotion in a simulated C. elegans nematode.

### Major Concerns

1. **Mathematical Rigor (Score: 4)**
   The paper introduces some formal elements: states are defined in terms of predicates, transitions between states are associated with probabilistic rules, and a utility function is mentioned for decision-making. However, these concepts are not developed mathematically. There are no definitions of the probability space, no learning algorithm with convergence guarantees, no formal treatment of the utility function, and no theoretical analysis of the proposed architecture. The formalization remains at a descriptive level.

2. **Novelty & Contribution (Score: 4)**
   The idea of drawing inspiration from biological theories of behavior for AI and robotics is not new. The authors themselves have published extensively on this topic (multiple self-citations). The mapping of TFS to CPS is a conceptual exercise that does not introduce new algorithms, models, or insights beyond the authors' previous work. The application to C. elegans is referenced from a 2014 paper, so no new experimental results are presented. The contribution is incremental at best.

3. **Relevance to MathAI (Score: 6)**
   The paper touches on topics relevant to the mathematics of AI: probabilistic rules, utility-based decision-making, learning from experience. However, the treatment is too shallow to be of interest to a mathematical audience. The connection to mathematics is mostly by allusion rather than substance.

4. **Technical Quality (Score: 4)**
   The methodology is described in general terms, with no technical details that would allow replication. The C. elegans application is mentioned only briefly, with no results, metrics, or comparisons. The paper does not stand alone as a technical contribution; it reads as an extended abstract or a position paper.

5. **Clarity & Presentation (Score: 6)**
   The paper is reasonably well-structured and the progression from TFS to CPS is logical. However, some sections are repetitive (e.g., the list of TFS steps appears twice with minor variations). The figures referenced (Fig. 2 and Fig. 3) are not visible in the text. The language is generally clear but could be more concise.

6. **AI-Generation Risk (Score: 2)**
   The paper appears to be human-written. It contains specific references to the authors' own work and to a well-defined theoretical framework (TFS). The style is consistent with academic writing in this niche area. No obvious signs of AI generation.

### Pros
- Interesting interdisciplinary perspective linking physiology and engineering.
- Attempts to provide a formal mapping between a biological theory and a computational architecture.
- Cites a concrete application (C. elegans simulation), suggesting some practical grounding.

### Cons
- Limited mathematical depth; no theorems, proofs, or rigorous analysis.
- Incremental novelty; heavily reliant on authors' prior work.
- No new experimental results or validation.
- Superficial treatment of key components (learning, utility, probabilistic rules).
- Does not meet the standards of a full research paper for a mathematics-focused conference.

### Recommendation
The paper presents an interesting conceptual framework but lacks the mathematical rigor, novelty, and technical depth expected at MathAI 2026. It would be more suitable for a workshop on cognitive architectures or bio-inspired computing. In its current form, it is not ready for acceptance.

---

### Official Review · Reviewer_G7Cq · 2026-03-13
**Anthropomorphic Cyber-Physical Systems**

**Rating:** 8
**Confidence:** 4

**Review:**

Summary:
This paper proposes an architectural framework for anthropomorphic cyber-physical systems inspired by the Theory of Functional Systems from neuroscience. The authors formalize CPS behavior in terms of sensor-based states, actions that produce probabilistic transitions, and goal predicates that define desired system states. Decision-making is organized hierarchically through functional subsystems responsible for goals and sub-goals, while learning is performed through probabilistic rule extraction from system experience. The framework integrates ideas from neuroscience, AI, and cyber-physical control systems.

Strengths:
- The paper presents an interesting biologically inspired perspective on the design of cyber-physical systems.
- The hierarchical organization of functional systems offers a structured way to model goal-directed behavior in CPS.
- The formalization of CPS states using predicates provides a clear conceptual framework for representing system behavior.
- The integration of probabilistic rules and reinforcement-based learning mechanisms aligns with modern approaches to adaptive control systems.
- The experimental illustration with a simulated C. elegans locomotion model demonstrates the feasibility of the approach.

Suggestions for improvement:
The paper could be strengthened by:
- providing a more detailed mathematical formulation of the learning and decision-making mechanisms;
- expanding the experimental evaluation with quantitative performance metrics;
- clarifying the relationship between the proposed architecture and existing AI planning or reinforcement learning frameworks.

Final Recommendation:
Acceptance with revision (the main strength is that the article shows a biologically inspired architecture for CPS and goal-directed AI)

Overall, the paper proposes an interesting biologically inspired architecture for cyber-physical systems that may stimulate discussion on goal-directed AI architectures and their applications to CPS.

---

### Decision · Program_Chairs · 2026-03-14

**Decision:**

Accept (Oral)

**Comment:**

Dear Author(s),

On behalf of the Program Committee of the International Conference on Mathematics of Artificial Intelligence (MathAI 2026), we are pleased to inform you that your paper has been accepted for an oral presentation at MathAI 2026.

Your paper was evaluated through a rigorous two-stage review process involving both automated screening and expert review by members of the Program Committee. The reviewers recognized the quality and contribution of your work.

Presentation details:

- Format: Oral presentation (15–20 minutes + 5 minutes Q&A)
- Mode: You may present either in person (offline) at the conference venue in Sirius, Russia, or remotely via Zoom. Please indicate your preferred mode when confirming your participation.
- Conference dates: Marh 30 - April 3, 2026
- Website: https://mathai.club

Next steps:

1. Please confirm your participation and presentation mode by replying to this email mathai.club@yandex.ru no later than March 15, 2026 18:00 Moscow time.
2. If you plan to attend in person, the organizing committee will provide accommodation details separately.
3. Please prepare your final camera-ready manuscript according to the formatting guidelines available at https://mathai.club and upload it to OpenReview by March 15, 2026 18:00 Moscow time.

Should you have any questions regarding the program, logistics, or your presentation slot, please do not hesitate to contact us.

We look forward to your contribution to MathAI 2026.

With kind regards,

MathAI 2026 Program Committee
International Conference on Mathematics of Artificial Intelligence
https://mathai.club
OpenReview: https://openreview.net/group?id=mathai.club/MathAI/2026/Conference
Telegram: https://t.me/MathAI_club
Email: mathai.club@yandex.ru